Exploring differential health effects of work stress: a latent class cluster approach

Mayerl Hannes hannes.mayerl@medunigraz.at
Stolz Erwin
Waxenegger Anja
Freidl Wolfgang
Institute of Social Medicine and Epidemiology, Medical University of Graz , Graz , Austria
Hung Tsung-Min
Electronic publication date: 2017 Mar 21
Publication date: 2017
Volume: 5
Electronic Location ID: e3106
Received 2016 Nov 30; Accepted 2017 Feb 20
Copyright: ©2017 Mayerl et al.
Copyright year: 2017
Copyright holder: Mayerl et al.
License: This is an open access article distributed under the terms of the Creative Commons Attribution License, which permits unrestricted use, distribution, reproduction and adaptation in any medium and for any purpose provided that it is properly attributed. For attribution, the original author(s), title, publication source (PeerJ) and either DOI or URL of the article must be cited.
License URL: https://creativecommons.org/licenses/by/4.0/

Keywords: Stress profiles, Symptom clusters, Latent class analysis, Person-centred approach

Funding: The authors received no funding for this work.

==============================
Background

While evidence highlights the detrimental health consequences of adverse working conditions, effect sizes vary by the stressor examined. In this study, we aimed to explore the differential effects various constellations of job demands have on prevalent symptom clusters.

Methods

We analysed self-reported data from a nationwide Austrian survey (N = 16,466), based on a cross-sectional design. By means of latent class analysis, a set of items was used to assess the burden from several job demands as well as the frequency of occurrence of mental and physical symptoms in order to identify stress profiles and symptom clusters, respectively.

Results

Analysis revealed four subgroups that each demonstrated a typological response pattern regarding job demands and health symptoms, respectively. The revealed stress profiles were found to be strongly related to the symptom clusters, while the effects differed considerably depending on the types of demands experienced.

Conclusion

The current study presents an alternative method of examining the stress-health link by using a combined person- and variable-centred approach. The findings suggest a hierarchy in stress exposure with the most pronounced health consequences found for a synchronous burden from physical, psychosocial and organizational demands.

Introduction

There is good empirical evidence that shows the negative impact of adverse working conditions on both mental and somatic health (Nixon et al., 2011; Siegrist, 2008). Previous research in this context either focused on individual health problems such as musculoskeletal symptoms (Mikkonen et al., 2012), cardiovascular diseases (Backé et al., 2012), gastrointestinal disorders (Suarez et al., 2010), or impaired sleep (Âkerstedt, 2006), or used a composite symptom score of diverse health problems (see e.g., Spector & Jex, 1998).

A critical point to consider when using composite symptom scores is that many occupational research questions require a homogenous population. However, this is rarely feasible since the working population usually consists of very heterogeneous groups of individuals. Such a heterogeneity may be directly observable (e.g., young vs. old persons) or, if not directly observable, be inferred from a set of measured indicator variables. For instance, the population may consist of different groups of individuals that vary in their vulnerability to specific stress-related health problems, i.e., some individuals might be more susceptible with respect to the digestive system, others might be more vulnerable in relation to the musculoskeletal system, and again others might be more likely to show psychological symptoms. More generally speaking, different groups of individuals may show distinct patterns of symptoms and health problems depending on their genetic predisposition, personality, health behaviours, resources, or life stories (McEwen, 1998).

Such an unobserved group membership can be represented by a categorical latent variable that can be estimated based on an appropriate set of manifest indicator variables. Therefore, rather than conceptualizing health problems as a continuous variable, an alternative—or complementary—way of looking at the data would be to form distinct subgroups of subjects with similar response patterns regarding the particular indicator variables. This so-called person-centred approach (in contrast to the variable-centred approach; see Muthén & Muthén, 2000) focuses on similarities or relationships among individuals (rather than relationships among variables) and can serve to reveal the unobserved heterogeneity within the population and to identify a typology of health problems within the working context. Burkert et al. (2015), for example, used both negative (e.g., psychosomatic ailments, back pain, irritation) and positive (e.g., self-efficacy, social orientation and participation, spiritual meaning) health indicators, in order to identify a health typology within a large sample of wage and salary earners. They revealed four distinct health clusters: (1) a cluster of physically healthy wage and salary earners with signs of mental issues such as demotivation and a limited desire to grow, (2) an overall healthy cluster (i.e., few health impairments and high levels of positive health indicators), (3) a cluster with a high tendency to wear out (i.e., increased levels of psychological impairments such as burnout symptoms), and (4) a cluster with multiple health issues and low levels of positive health indicators.

Not only in terms of health consequences but also in terms of potential causes (i.e., stressors and demands), research has found a structuring within the population (Vanroelen et al., 2010). In the world of work, individuals encounter various forms of exposure to a theoretically infinite set of job demands and stressors, and current stress research shows special interest in structuring the perceived burden due to such demands in order to identify different stress profiles (e.g., Jenull & Wiedermann, 2015). For example, Vanroelen et al. (2010) imposed a taxonomy upon wage-earners, based on a set of scales assessing quantitative (e.g., time pressure) and emotional (e.g., contact with difficult clients) demands, as well as physical, social, and organizational working conditions. Vanroelen et al. (2010) identified five different subgroups: (1) a low stress cluster showing a generally low burden due to job demands and working conditions, (2) a passive-manual cluster characterized by a burden from aspects such as adverse physical working conditions, repetitive movements, or low job control, (3) a human contact cluster reporting stress especially due to emotional demands, or problematic relationships with superiors, (4) a high stress cluster distinguished by a generally high risk for demands and stressors of any kind, (5) a high demand cluster that was predominantly characterized by a high risk for quantitative and emotional demands, and a low risk for adverse physical working conditions and detrimental organizational job aspects such as atypical work schedule or low job control. Vanroelen et al. (2010) also found that each of these stress clusters demonstrated distinct associations with emotional problems and musculoskeletal complaints.

The research of Vanroelen et al. (2010) thus not only supports the existence of different clusters showing distinct patterns of stress exposure but also stresses the importance of considering this heterogeneity when examining the relationship between stress exposure and health. That is, health consequences can vary considerably depending on the types of demands to which individuals are exposed. This finding is in line with the results of a meta-analysis (Nixon et al., 2011); it showed, among other things, that the stressor interpersonal conflict had a stronger effect (weighted effect size) on sleep disturbances (.22) than on fatigue (.09) and that workload was more strongly related to fatigue (.31) than to sleep disturbances (.14). Since various single demands or combinations of demands can affect health in very different ways, the widely used approach of examining the stress-health link on the basis of composite scores may lead to unreliable results.

Therefore, stress research should account for (1) potential heterogeneity in the vulnerability to health problems, (2) a structuring in the perceived burden due to job demands, and (3) the varying impact of different types of demands on health. To our knowledge, however, the relationship between different stress profiles and symptom clusters has never been examined to date. We suggest that such an approach may shed new light on the complex associations between work-related stress and health, while including a broad set of both health indicators and job demands on the one hand, and accounting for the potential heterogeneity in both the outcome and the explanatory variable on the other hand. Thus, the major objective of our study was twofold: first of all, this study aimed to reveal both different profiles regarding the perceived burden due to job demands, and different clusters in terms of the occurrence of various health symptoms. Second, we aimed to predict the revealed symptom clusters on the basis of the revealed stress profiles in order to explore differential health effects of various combinations of demands.

The current study

Our analysis consisted of two main steps: In the first step, we aimed to identify subgroups of individuals that showed similar response patterns in the respective indicator variables, in order to explore different stress profiles and symptom clusters. The indicator variables for health problems referred to a broad set of somatic symptoms regarding the digestive system, the musculoskeletal system, the respiratory system, the circulatory system, skin tissue as well as mental health. Job demands were operationalized by a heterogeneous set of items comprised of “physical, psychological, social, or organizational aspects of the job that require sustained physical and/or psychological (cognitive and emotional) effort and are therefore associated with certain physiological and/or psychological costs” (Bakker et al., 2003, p. 344).

In the second step, we attempted to predict latent group membership based on symptom clusters via the latent group membership based on stress profiles. To achieve this, we categorized each subject into the most likely subgroup revealed in step one and then examined the relationship between the two resulting latent variables. Since both the first and the second step were of an exploratory nature, no specific hypotheses about the number of different stress profiles and symptom clusters, their characteristic patterns, and their relationship were made. However, we assumed a heterogenic population in terms of both health problems and job demands, and therefore hypothesized at least two subgroups for each latent variable. Moreover, we expected the likelihood of belonging to a specific symptom cluster to vary according to the assigned stress profile.

Materials & Methods

Data

This study was based on data extracted from a representative nationwide survey conducted for the Austrian Work Climate Index and the Austrian Employee Health Monitor by the Institute for Empirical Social Studies (IFES) on behalf of the Upper Austrian Chamber of Labour. Data were collected between 2009 and 2014 among the Austrian working population using proportionally stratified random sampling. The IFES conducted structured face-to-face interviews at participants’ homes to gather self-reported data concerning working conditions, health status, and demographics. Since the relevant data were only available for employed persons, we excluded self-employed persons and liberal professions. We further excluded all persons for which more than one third of the data was missing (15 cases in total). The resulting sample size was N = 16, 466 and the proportion of missing values among the used data set was a mere 0.70%.

Ethics statement

Data collection was carried out in compliance with the principles defined in the Helsinki Declaration. Trained interviewers informed participants on the confidentiality and anonymity of the collected data and stated the study objectives. All interviews were conducted after verbal informed consent. In the case of 15–17 year-olds, the legal guardian had to agree. The Ethics Committee of the Medical University of Graz approved the conductance of the current study (Ethical Application Ref: 27-251 ex 14/15).

Indicator variables

Health symptoms

We used 16 items concerning somatic and mental health issues. Participants had to indicate on a 5-point rating scale, ranging from 1 = never to 5 = very often, how often in the last weeks they had suffered from: (1) digestive problems, (2) stomach troubles, (3) migraine/headaches, (4) difficulties falling or remaining asleep, (5) exhaustion/weariness/depressiveness, (6) nervousness/absent-mindedness, (7) weakness of memory/lack of concentration, (8) muscle tenseness in neck and shoulder regions, (9) back pain, (10) leg pain, (11) hypertension, (12) tachycardia/palpitations/feeling of pressure on the chest, (13) skin rash/itching/skin redness, (14) respiratory problems/shortness of breath/breathlessness/asthma, (15) a chronic cough, or (16) eye problems (burning, itching, inflammation, allergies). These variables were chosen because they have been found to be associated with work-related stress reactions (Nixon et al., 2011) and because comparable sets of items are commonly used in large-scale health studies (Zijlema et al., 2013).

Job demands

To assess the subjectively perceived burden from physical, mental, social, and organizational job demands, we used up to five indicator variables for each of these domains, in order to obtain a reliable measure and to cover a broad spectrum of potential stressors at work. Items which were either redundant or too specific were excluded. This resulted in 19 items in total. Participants were asked to indicate the extent to which they felt burdened by (1) standing activity/forced posture, (2) heavy physical exertion/heavy lifting tasks, (3) one-sided physical strain, (4) frequent computer work, (5) isolation at the workplace, (6) time pressure, (7) emotionally burdening and annoying work, (8) a lack of opportunity to retreat, (9) constant demand to be highly concentrated, (10) high responsibility for goods and people, (11) permanent monitoring and surveillance, (12) permanent contact to customers/clients, (13) permanent contact with seriously ill or needy persons, (14) lack of support from the supervisor, (15) lack of support from colleagues, (16) technical and organizational changes, (17) changes to work routines, (18) irregular working hours, and (19) excessive working hours. Response categories ranged from 1 = not burdened to 5 = strongly burdened.

Covariates

Job resources

We defined job resources as all aspects of a job that relate to any of the following: (a) are functional in goal achievement, (b) diminish job demands and the associated costs, and/or (c) facilitate personal growth and development (Bakker et al., 2003; Demerouti et al., 2001). We included a total of six items to assess the level of satisfaction with (1) opportunities to autonomously decide about work processes, (2) opportunities for co-determination at workplace, (3) working rights, (4) occupational training opportunities, (5) career and development opportunities, and (6) income. Participants had to rate these items on a 5-point scale ranging from 1 = not at all satisfied to 5 = very satisfied. We used the mean score of these items in subsequent analyses. Internal consistency of this scale was satisfactory (Cronbach’s α = 0.88).

Self-efficacy

General self-efficacy has been defined as a global confidence in one’s ability to cope with a wide range of challenging or novel situations (Schwarzer et al., 1997). A general construct of self-efficacy was found to be more appropriate in predicting broad outcomes such as well-being or health, compared to more specific constructs of self-efficacy (Schwarzer et al., 1997; Schwarzer & Hallum, 2008). General self-efficacy was measured using three items (e.g., “I am confident that I could deal efficiently with unexpected events”.) from the German version of the General Self-efficacy Scale (Schwarzer & Jerusalem, 1995; Schwarzer & Jerusalem, 1999). Response categories ranged from 1 = do not agree to 5 = strongly agree. The original General Self-efficacy Scale (consisting of ten items in total) was previously found to show good reliability and correlations with depression, anxiety, or optimism confirmed construct validity (Schwarzer et al., 1997). Again, we used the mean score of the three items. Internal consistency was very good (Cronbach’s α = 0.88).

Educational level

This variable reflects the highest level of education reached, classified into four groups: compulsory school/skilled workers holding an apprenticeship certificate or a diploma from vocational school/high school diploma/university degree.

Occupational groups

We classified occupations into nine groups based on the International Standard Classification of Occupations (ISCO-08): “1 Managers”, “2 Professionals”, “3 Technicians and associate professionals”, “4 Clerical support workers”, “5 Service and sales workers”, “7 Craft and related trades workers”, “8 Plant and machine operators, and assemblers”, “9 Elementary occupations”, and the residual category “Undefined”. Due to the low frequency of occurrence, we excluded “0 Armed forces occupations” (N = 60) and “6 Skilled agricultural, forestry and fishery workers” (N = 78) from all analyses including occupational groups.

Statistical analysis and procedure

We applied latent class analysis (LCA) on health symptoms and job demands to identify different stress profiles and symptom clusters, respectively. Since the number of latent classes cannot be estimated as part of the LCA, we evaluated models with up to seven classes and defined the number of classes based on statistical and substantive grounds (see also Lawrence & Zyphur, 2011; Marsh et al., 2009). For the purpose of statistical model selection, we used the Bayesian information criterion (BIC; Schwarz, 1978), which was found to be a useful indicator for defining the number of latent classes (Nylund, Asparouhov & Muthén, 2007). Furthermore, we considered the relative improvement in model fit (based on the log-likelihood-function) between the k-class and the (k + 1)-class model (Bacher & Vermunt, 2010). As a criterion for classification quality, we used the entropy measure (Celeux & Soromenho, 1996), which ranges between 0 and 1, where values greater than 0.80 indicate good class separation (SL Clark & BO Muthén, 2009, unpublished data). In addition to the statistical model fit assessment, we evaluated the competing models in terms of usefulness and interpretability of the posterior item probability profiles.

Even though latent class software packages can handle Likert-type data of ordinal measurement scale, we decided to dichotomize the data prior to conducting LCA for two main reasons: First, since we included a relatively high number of indicator variables, a five-point rating scale would have resulted in an extensive number of possible response patterns, while a considerable proportion of these patterns was not or infrequently observed in the sample. This may lead boundary parameter estimates (probabilities estimated to be zero or one) to cause severe problems in the estimation process (Galindo-Garre & Vermunt, 2006; Vermunt & Magidson, 2004; Wurpts & Geiser, 2014).

Second, the use of a higher number of response categories makes the interpretation of the item probability profiles more difficult and less intuitive. Therefore, we combined the response categories for each indicator variable from 1 to 3 and from 4 to 5 into one category, respectively. The rationale behind the use of the particular collapsing strategy in our study was that this would help us to obtain a better distinction between slightly to moderately affected cases and more severe cases. This approach is similar to the commonly used approach in health research where participants’ self-rated health is categorized as fair or poor versus moderate, good or excellent (see e.g., Ferrie et al., 2002; Marmot et al., 1998; Wolff et al., 2010; Zajacova & Dowd, 2011). For an extensive methodological discussion about the practice of dichotomization please refer to MacCallum et al. (2002).

Having found a well-fitting and useful latent class solution for both health symptoms and job demands, we classified each case according to its most likely class based on the posterior probabilities for latent class membership. This allowed us to obtain two categorical latent variables that reflected each person’s most likely latent class membership. In a second step, we estimated a multinomial logistic regression model by having the latent variable for symptom clusters regress on the latent variable for stress profiles. Moreover, given that job resources and self-efficacy were found to show substantial relationships to both work-related stress and health (see e.g., Mayerl et al., 2016; Xanthopoulou et al., 2007), we controlled for their confounding effects by including them as covariates in the final regression model. Additionally, we considered gender, age, education, and occupational groups.

All statistical analyses were performed using R 3.1.2 (R Core Team, 2014), while LCA and multinomial logistic regression analysis were accomplished with the R-packages poLCA 1.4.1 (Linzer & Lewis, 2011) and nnet 7.3-8 (Venables & Ripley, 2003), respectively. Missing values were imputed using random forest imputation from the R-package MissForest 1.4 (Stekhoven & Bühlmann, 2012), which was recently found to outperform more conventional methods such as multiple imputation, in terms of accuracy and efficiency (Liao et al., 2014; Stekhoven & Bühlmann, 2012; Waljee et al., 2013). Anonymized raw data and the R code are accessible via https://osf.io/a76wh/.

Table 1 Model selection criteria of the seven models.

LL, Log-likelihood; PV1k = ((−2⋅LLk−1) − (−2⋅LLk))∕(−2⋅LLk)⋅100; df, degrees of freedom; BIC, Bayes Information Criterion; ΔBIC, Difference in the BIC between the k − 1 and the k class model.

Classes	Symptom clusters	Stress profiles	
	LL	PV1	df	BIC	ΔBIC	Entropy	LL	PV1	df	BIC	ΔBIC	Entropy	
1	−48831.78	–	16	97818.90	–	–	−106044.23	–	19	212272.93	–	–	
2	−42108.07	13.77	33	84536.54	13282.36	0.83	−91237.62	13.96	39	182853.89	29419.03	0.85	
3	−41113.47	2.36	50	82712.40	1824.14	0.80	−89043.05	2.41	59	178658.94	4194.96	0.86	
4	−40828.79	0.69	67	82308.08	404.32	0.81	−87361.57	1.89	79	175490.16	3168.78	0.83	
5	−40732.79	0.24	84	82281.15	26.93	0.79	−86813.47	0.63	99	174588.14	902.02	0.82	
6	−40670.27	0.15	101	82321.15	−40.00	0.79	−86316.12	0.57	119	173787.62	800.52	0.80	
7	−40611.29	0.15	118	82368.25	−47.10	0.80	−85970.65	0.40	139	173290.86	496.76	0.81	

Figure 1 Conditional item probabilities of health symptoms for the 4-class model.

Results

Symptom clusters

Model fit criteria for the different class solutions regarding symptom clusters are summarized in Table 1. According to PV1, the likelihood increases only slightly from the 3-class model to the 7-class model and the BIC values point to a 4- or 5-class solution. The entropy measure indicates good classification quality for all models. In further evaluating the eligible models, we found the 4-class model to allow an adequate representation of the data and to permit good differentiation of the posterior probability profiles.

The conditional item probabilities of the 4-class solution are shown in Fig. 1. The first class had an estimated population size of 75.0% and the likelihood of (very) frequently suffering from any health symptoms was generally low within this class. The second-largest class (13.9%) was characterized by a high probability for back pain, muscle tenseness in neck and shoulder regions, and a slightly to moderately increased probability of having leg pain, migraine/headaches, difficulties falling or remaining asleep, exhaustion/weariness/depressiveness, and hypertension. The third class had a size of 8.3% and showed a moderately increased probability of having problems with exhaustion/weariness/depressiveness, muscle tenseness in neck and shoulder regions, difficulties falling or remaining asleep, and migraine/headaches, as well as a slightly increased probability for all other health symptoms. The fourth class had a relative size of 2.8% and showed the highest likelihood for all health symptoms, in particular for muscle tenseness in neck and shoulder regions, back pain, exhaustion/weariness/depressiveness, difficulties falling or remaining asleep, and nervousness/absent-mindedness. Based on these patterns, we identified the classes in descending order of class size as healthy, tensed up, exhausted, and heavily suffering.

The distribution of the latent class probabilities over gender, age, education, and the occupational groups can be found in Table 2. There was a clear gender effect, in that women (vs. men) were less likely to be classified as healthy and more likely to be categorized into the classes with health symptoms. Furthermore, the chances of belonging to the healthy class were higher in young age and lower in old age, whereas, generally speaking, young employees were less and old employees were more likely to be put into the “unhealthy” categories. Although education and occupational groups were found to be significantly associated with the symptom clusters, there was less evidence for systematic health inequalities among different levels of education or occupational groups.

Table 2 Distribution of the latent class probabilities in percent for the symptom clusters over gender, age, education, and occupational groups.

All χ2-tests of independence were significant (all ps <.001). In parentheses, we report the standardized residuals of the respective independence tests. These values result when calculating the difference between the observed and expected frequencies, divided by its standard error (Agresti, 2007). Standardized residuals exceeding (or going below) the critical value on level α = .01 divided by the number of cells (Bonferroni-correction) indicate a significantly larger (or lower) cell frequency as expected by chance.

	N	Symptom cluster	
		Healthy	Tensed up	Exhausted	Heavily suffering	
Overall	16,328	79.20	12.23	5.93	2.64	
Gender						
Men	8,069	82.19 (9.29*)	11.05 (−4.40*)	4.93 (−5.51*)	1.82 (−6.40*)	
Women	8,259	76.29 (−9.29*)	13.31 (4.40*)	6.97 (5.51*)	3.43 (6.40*)	
Age						
15–29	3,852	86.47 (12.71*)	6.62 (−12.09*)	4.96 (−3.02*)	1.95 (−3.04*)	
30–49	8,753	79.64 (1.47)	11.65 (−2.27)	6.23 (1.51)	2.48 (−1.32)	
≥50 years	3,723	70.67 (−14.61*)	19.23 (14.94*)	6.39 (1.25)	3.71 (4.65*)	
Education						
Compulsory school	1,518	72.53 (−6.73*)	16.80 (5.76*)	6.65 (1.19)	4.02 (3.54*)	
Skilled workers/vocational school	10,456	79.69 (2.01)	12.65 (2.39)	5.22 (−5.35*)	2.44 (−2.07)	
High school	2,263	82.28 (3.88*)	9.46 (−4.29*)	6.19 (0.48)	2.08 (−1.78)	
University	2,091	78.34 (−1.05)	9.52 (−4.01*)	8.94 (6.16*)	3.20 (1.75)	
Occupational groups (ISCO-08)						
1 Managers	462	80.09 (0.47)	9.31 (−1.92)	6.49 (0.49)	4.11 (2.01)	
2 Professionals	1,937	77.39 (−2.10)	10.79 (−2.01)	8.72 (5.46*)	3.10 (1.36)	
3 Technicians/associate professionals	2,543	78.10 (−1.50)	12.94 (1.25)	5.98 (0.03)	2.99 (1.22)	
4 Clerical support workers	2,467	83.62 (5.87*)	8.51 (−6.07*)	5.72 (−0.57)	2.15 (−1.63)	
5 Service and sales workers	3,325	81.29 (3.32*)	11.70 (−0.98)	4.66 (−3.56*)	2.35 (−1.16)	
7 Craft and related trades workers	1,740	83.28 (4.42*)	11.90 (−0.40)	3.05 (−5.44*)	1.78 (−2.35)	
8 Plant and machine operators	550	76.55 (−1.56)	16.36 (3.04)	4.73 (−1.25)	2.36 (−0.40)	
9 Elementary occupations	1,909	74.49 (−5.41*)	17.44 (7.46*)	5.03 (−1.84)	3.04 (1.18)	
Undefined	1,395	73.12 (−5.86*)	12.97 (0.93)	10.90 (8.13*)	3.01 (0.92)	
Notes.

* indicates significance.

ISCO-08 International Standard Classification of Occupations

Figure 2 Conditional item probabilities of job demands for the 4-class model.

Stress profiles

As for job demands, PV1 indicated that the increase in model fit was only minor for the 4- to 7-class models. Also the ΔBICs showed less improvement from the 4-class model upwards. Again, entropy measures were very good for all seven models. Thus, we further evaluated the models with 4–7 latent classes in terms of interpretability and usefulness, and found that the 4-class model permitted an adequate and meaningful representation of the data while not being over- or underfitted.

Table 3 Distribution of the latent class probabilities in percent for the stress profiles over gender, age, education, and occupational groups.

All χ2-tests of independence were significant (all ps <.001). In parentheses, we report the standardized residuals of the respective independence tests.

	N	Stress profile	
		Low burden	Psychosocial burden	Physical burden	High burden	
Overall	16,328	59.03	22.43	10.15	8.39	
Gender						
Men	8,069	57.63 (−3.69*)	21.14 (−3.95*)	12.36 (9.54*)	8.87 (2.11)	
Women	8,259	60.47 (3.69*)	23.72 (3.95*)	7.86 (−9.54*)	7.95 (−2.11)	
Age						
15–29	3,852	63.55 (6.48*)	19.42 (−5.15*)	9.22 (−2.04)	7.81 (−1.52)	
30–49	8,753	58.61 (−1.27)	22.92 (1.55)	9.99 (−0.44)	8.49 (0.39)	
≥50 years	3,723	55.49 (−5.04*)	24.47 (3.37*)	11.20 (2.58)	8.84 (1.07)	
Education						
Compulsory school	1,518	54.74 (−3.60*)	11.26 (−10.96*)	22.33 (16.65*)	11.66 (4.79*)	
Skilled workers/vocational school	10,456	61.22 (7.47*)	19.60 (−11.65*)	11.28 (6.77*)	7.91 (−3.07*)	
High school	2,263	61.73 (2.78)	28.90 (7.93*)	3.45 (−11.29*)	5.92 (−4.59*)	
University	2,091	48.54 (−10.48*)	37.83 (18.05*)	2.39 (−12.51*)	11.24 (4.99*)	
Occupational groups (ISCO-08)						
1 Managers	462	55.41 (−1.62)	31.82 (4.90*)	3.46 (−4.79*)	9.31 (0.71)	
2 Professionals	1,937	51.57 (−7.14*)	36.09 (15.33*)	2.99 (−11.03*)	9.34 (1.58)	
3 Technicians/associate professionals	2,543	48.84 (−11.41*)	29.37 (9.11*)	9.36 (−1.32)	12.43 (7.94*)	
4 Clerical support workers	2,467	71.02 (13.10*)	24.73 (2.95)	1.01 (−16.24*)	3.24 (−10.04*)	
5 Service and sales workers	3,325	66.08 (9.21*)	16.93 (−8.54*)	9.44 (−1.37)	7.55 (−2.00)	
7 Craft and related trades workers	1,740	58.62 (−0.40)	11.84 (−11.22*)	22.47 (18.16*)	7.07 (−2.13)	
8 Plant and machine operators	550	39.27 (−9.60*)	31.27 (5.05*)	12.91 (2.24)	16.55 (6.99*)	
9 Elementary occupations	1,909	57.26 (−1.71)	12.10 (−11.53*)	21.22 (17.19*)	9.43 (1.71)	
Undefined	1,395	62.29 (2.57)	20.79 (−1.55)	9.18 (−1.17)	7.74 (−0.94)	
Notes.

* indicates significance after Bonferroni-adjustment.

ISCO-08 International Standard Classification of Occupations

The conditional item probabilities of the 4-class solution are illustrated in Fig. 2. The largest class (57.8%) had an overall low probability to be burdened by any of the job demands. The second class had a prevalence of 23.1% and was characterized by a moderate to high probability of feeling burdened by psychosocial demands (especially high responsibility for goods and people, constant demand to be highly concentrated, contact to customers/clients, time pressure) or burdened by computer work, and a slightly to moderately increased probability of feeling burdened by emotionally burdening and annoying work, or by a lack of opportunity to retreat. 10.4% of the employed persons are expected to belong to the third class that showed the highest probabilities in terms of a burden due to physical demands (one-sided physical strain, heavy physical exertion/heavy lifting tasks, standing activity/forced posture), and a moderately to high probability for a burden due to time pressure, high responsibility for goods and people, and constant demand to be highly concentrated. Class 4 had a size of 8.7% and showed a generally high probability to be burdened by any demands (except for the burden due to isolation at the workplace, which was generally low in all four classes). Based on the profiles, the labels for the classes were as follows (in descending order of class size): low burden, psychosocial burden, physical burden, and high burden.

The distribution of the latent class probabilities over gender, age, education, and occupational groups is shown in Table 3. Women (vs. men) were more likely to belong to the low burden class and the psychosocial burden class and less likely to belong to the physical burden class. Moreover, young employees were more likely to be in the low burden class and less likely to suffer from a psychosocial burden. Older employees, in turn, were less likely to be classified as low burdened and more likely to belong to the psychosocial burden class. Higher levels of education were more strongly associated with a psychosocial burden and lower levels of education with a physical burden. Employees with compulsory school education and university graduates were less likely to belong to the low burden class, while skilled workers or those with vocational education had a higher likelihood of being in this class. By contrast, employees with at most compulsory school education or with a university graduation were more likely classified as highly burdened, while high school graduates and skilled workers or those with vocational education were less likely found in the high burden class. Again, occupational groups were significantly related to stress profiles, whereas no systematic pattern was evident for these groups.

Table 4 Multinomial logistic regression model for symptom cluster.

Job demands, gender, age, education, and ISCO-08 were included as dummy variables, while job resources and self-efficacy represent standardized measures. Pseudo R2 (Cox & Snell, 1989; McFadden, 1973; Nagelkerke, 1991): Cox & Snell, 0.17; McFadden, 0.13; Nagelkerke, 0.22.

	Symptom cluster (ref.: healthy)	
	Tensed up	Exhausted	Heavily suffering	
	β	OR [99% CI]	β	OR [99% CI]	β	OR [99% CI]	
(Intercept)	−3.43*	0.0 [0.0, 0.0]	−4.04*	0.0 [0.0, 0.0]	−5.95*	0.0 [0.0, 0.0]	
Stress profile (ref: low burden)							
Psychosocial burden	0.84*	2.3 [1.9, 2.7]	1.13*	3.1 [2.5, 3.9]	1.42*	4.1 [2.8, 6.2]	
Physical burden	1.54*	4.7 [3.8, 5.7]	0.98*	2.7 [1.9, 3.7]	2.22*	9.2 [5.8, 14.5]	
High burden	1.85*	6.4 [5.2, 7.9]	1.91*	6.7 [5.1, 8.9]	3.22*	25.0 [16.5, 37.7]	
Job resources	−0.34*	0.7 [0.7, 0.8]	−0.29*	0.8 [0.7, 0.8]	−0.50*	0.6 [0.5, 0.7]	
Self efficacy	−0.01	1.0 [0.9, 1.1]	−0.21*	0.8 [0.7, 0.9]	−0.04	1.0 [0.8, 1.1]	
Gender (ref: Men)							
Women	0.56*	1.7 [1.5, 2.0]	0.49*	1.6 [1.3, 2.0]	1.00*	2.7 [2.0, 3.7]	
Age (ref: 15–29)							
30–49	0.66*	1.9 [1.6, 2.4]	0.29*	1.3 [1.1, 1.7]	0.36*	1.4 [1.0, 2.1]	
≥50 years	1.36*	3.9 [3.2, 4.8]	0.48*	1.6 [1.2, 2.1]	1.00*	2.7 [1.8, 4.1]	
Education (ref: compulsory school)							
Skilled workers/vocational school	0.05	1.1 [0.8, 1.3]	−0.06	0.9 [0.7, 1.3]	−0.11	0.9 [0.6, 1.4]	
High school	−0.13	0.9 [0.6, 1.2]	−0.05	0.9 [0.6, 1.4]	−0.31	0.7 [0.4, 1.3]	
University	−0.43*	0.6 [0.5, 0.9]	−0.00	1.0 [0.7, 1.5]	−0.31	0.7 [0.4, 1.4]	
Occupational groups (ref: 9 elementary)							
1 Managers	−0.12	0.9 [0.5, 1.4]	0.54	1.7 [0.9, 3.1]	1.05*	2.9 [1.3, 6.3]	
2 Professionals	0.00	1.0 [0.7, 1.4]	0.56*	1.8 [1.1, 2.7]	0.51	1.7 [0.9, 3.1]	
3 Technicians/associate professionals	−0.13	0.9 [0.7, 1.1]	0.20	1.2 [0.8, 1.8]	0.21	1.2 [0.7, 2.1]	
4 Clerical support workers	−0.29*	0.7 [0.6, 1.0]	0.31	1.4 [0.9, 2.0]	0.40	1.5 [0.8, 2.7]	
5 Service and sales workers	−0.24*	0.8 [0.6, 1.0]	−0.10	0.9 [0.6, 1.3]	−0.10	0.9 [0.5, 1.5]	
7 Craft and related trades workers	−0.14	0.9 [0.7, 1.2]	−0.22	0.8 [0.5, 1.3]	0.06	1.1 [0.6, 2.0]	
8 Plant and machine operators	−0.11	0.9 [0.6, 1.3]	−0.17	0.8 [0.5, 1.5]	−0.16	0.9 [0.4, 2.0]	
Undefined	0.21	1.2 [0.9, 1.6]	1.14*	3.1 [2.1, 4.6]	0.70*	2.0 [1.1, 3.6]	
Notes.

* p < .01.

ISCO-08 International Standard Classification of Occupations.

ref. reference group

The relationship between stress profiles and symptom clusters

The results of the multinomial logistic regression model are shown in Table 4. All three classes in terms of stress profiles had, in comparison to the low burden class, a significantly higher chance of belonging to the tensed up, the exhausted, or the heavily suffering class than to the healthy class, while overall, the strongest effects were found for the high burden class. The odds of belonging to the healthy class (in comparison to the other three classes), in turn, increased as job resources increased. Self-efficacy was only related to the exhausted class, in that increasing levels of self-efficacy went along with a decline in the odds of belonging to the exhausted rather than to the healthy class. Moreover, women (vs. men) were more likely to be tensed up, exhausted, or heavily suffering. As for age, the odds of belonging to the “unhealthy” classes were higher among the middle and the older age groups than among young employees. In terms of education, only one significant result was found: university graduates were less likely to be classified as tensed up. On the level of occupational groups, an increased chance of belonging to the exhausted class or to the heavily suffering class was found for managers, professionals, and the residual category, while clerical support workers, and service and sales workers had a lower chance of being tensed up.

Discussion

The primary objectives of this research were to identify subgroups of individuals that demonstrated a typological pattern with regard to both the perceived burden due to different kinds of job demands and the vulnerability to health problems, and to reveal stress profiles that showed the greatest risk in terms of manifesting different constellations of health symptoms. Although the effect sizes varied considerably, we found strong associations between the profiles characterized by a psychosocial, physical or generally high burden and the symptom clusters demonstrating both somatic and mental health issues.

Symptom clusters

On the level of health problems, three quarters of the employees (i.e., the healthy class) were found to be relatively healthy as they indicated a low frequency of health symptoms, whereas the remainder, subdivided in three different classes, was found to (very) often suffer from specific health issues. Within this latter part, the tensed up symptom cluster represented the greatest population share and was predominantly characterized by a very high frequency of musculoskeletal symptoms such as problems in back, neck and shoulder regions. This finding conforms to previous studies, which had shown back and neck problems to become increasingly important public health issues (Fejer, Kyvik & Hartvigsen, 2005; Freburger et al., 2009; Großchädl et al., 2015).

The third-largest class (i.e., the exhausted symptom cluster) demonstrated moderately increased chances of typical work-related stress symptoms (see e.g., Nixon et al., 2011) such as exhaustion or depressiveness, sleeping problems, headache, and muscle tension, whereas this class can also be considered as a less pronounced version of the heavily suffering symptom cluster that exhibits a very critical health status with the highest probabilities for all health symptoms across all classes. Even though this latter class included a relatively small share of the population, this group merits special attention when it comes to health promotion and prevention policies.

In all classes, except for the healthy class, a consistent pattern regarding the co-occurrence of musculoskeletal problems—such as neck or back pain—and mental or emotional issues—such as exhaustion/depressiveness, sleeping problems or nervousness—became evident. Although such a co-morbidity has been reported in previous studies (e.g., Demyttenaere et al., 2007; Polatin et al., 1993), it is still interesting to know, particularly for the purpose of prevention, whether this co-occurrence is due to a common cause or to a causal relationship, i.e., whether mental issues cause neck and back problems or vice versa (Gatchel et al., 2007; Polatin et al., 1993). A prospective cohort study by Jarvik et al. (2005) found patients with depression at baseline to be 2.3 times more likely to develop back pain three years later than those who did not report any baseline depression. However, there is also evidence for a causal link between back pain and mental disorders. Compared to a control group, men with chronic back pain had higher lifetime prevalences for depression, alcohol use, or anxiety disorders (Atkinson et al., 1991). Future studies should, therefore, also consider factors that are supposed to mediate the link between musculoskeletal problems and mental health issues (e.g., cognitive appraisal; Turk, Okifuji & Scharff, 1995) in order to clarify the underlying mechanisms of this relationship.

Stress profiles

In regard to challenging working conditions, about 42% of the employees had an increased likelihood of feeling (heavily) burdened by specific patterns of job demands. Especially working environments that were experienced as psychosocially burdening (i.e., demonstrating the psychosocial stress profile) showed a relatively high prevalence. Within this class, factors associated with cognitive (e.g., high concentration) and psychosocial (e.g., high responsibility for goods and people) demands seemed to be particularly stressful, whereas only a minor or even no burden at all was found to result from excessive physical demands. The burden due to physically strenuous tasks, in contrast, most likely occurred within the third-largest class (i.e., the physical stress profile). This class, however, was not limited to physical factors alone, since the burden from psychosocial aspects such as pressure to meet deadlines also had a moderately high probability of occurrence. Most critically, almost 9% of employees were found to be at high risk of feeling burdened by both psychosocial and physical demands, as well as by organizational job aspects (i.e., demonstrating the high stress profile).

The revealed stress profiles are well comparable to those reported in Vanroelen et al. (2010). In accordance with our study, Vanroelen et al. (2010) revealed (1) a low stress cluster showing only slight burden due to work stressors, (2) a psychosocially burdened cluster reporting stress due to emotional or social factors in particular, (3) a physically burdened cluster characterized by a burden especially from physical demands, and (4) a highly stressed cluster showing a generally high burden due to work. In contrast to our study, however, Vanroelen et al. (2010) additionally revealed a fifth high demand cluster. The reason for this divergence is likely due to the use of considerably different indicator variables in these two studies. While Vanroelen et al. (2010) included a set of scales assessing not only job demands but also job resources (such as job control) in their latent class model, our study focused on individual job demands on the item level and included measures of both job resources and personal resources as covariates in the final regression model rather than in the latent class model. Nevertheless, the close correspondence in the results of the two studies supports the validity of the findings.

Health effects of job demands

When considering the relationship with symptom clusters, there seemed to be a job demands hierarchy depending on the seriousness of health consequences. Although a psychosocially burdening working environment appeared to be least severe, it should be kept in mind that workers within this class were nevertheless up to 4 times more likely (than the low burden class) to be classified as tensed up, exhausted, or heavily suffering. Our results also showed psychosocial stressors not only to be related to mental but also to somatic symptoms, i.e., a psychosocially burdening working environment was associated with musculoskeletal problems, even though the burden due to physical demands was only a minor one. This finding is in line with biopsychosocial models of pain that recognize psychosocial factors, such as chronic psychosocial stress, to be important in the aetiology, maintenance, or treatment of pain (Gatchel et al., 2007). Against this backdrop, psychosocial stress and the numerous related physiological responses (e.g., autonomic, immune, neuroendocrine responses) as well as potentially related behavioural changes (e.g., in terms of exercise, diet, smoking, drinking) are considered to be critical factors leading to health problems and pain perceptions (Gatchel et al., 2007; McEwen, 1998; McEwen, 2008).

On the other hand, predominantly physically burdened workers vs. psychosocially burdened workers, showed an even greater likelihood to be classified as tensed up or heavily suffering, while the odds of belonging to the exhausted class were of similar magnitude in both groups. Generally speaking, this suggests that high physical demands, even though sometimes combined with moderate psychosocial demands, may more often result in health problems than a purely psychosocially burdening working environment. Since current work-related stress research mainly focuses on psychosocial stressors (see e.g., Nixon et al., 2011), this study stresses that physical aspects and their combinations with psychosocial factors should be also considered.

The overall heavily burdened class even showed stronger associations with health, with an approximately 6–7 times increased chance of being categorized as tensed up or exhausted and a 25 times higher chance of belonging to the heavily suffering class than to the healthy class. These findings suggest that overall poor working conditions—including physical, mental, social, and organizational aspects of a job—may cumulate to reinforce negative effects on health. This can be explained by the allostatic load theory (McEwen & Stellar, 1993; McEwen, 1998; McEwen, 2008), which states that the human organism permanently adapts its internal physiological processes to meet the demands of the environment. The exposure to high demands over an extended period of time, however, causes the wear and tear of chronically activated physiological processes to damage the affected system. Thus, detrimental health effects may build up through cumulative stress exposure due to synchronous over-activated physiological processes. A further explanation is provided by the conservation of resources theory (Hobfoll, 1989), which considers individuals to have an innate drive to accumulate and protect resources (e.g., self-esteem, motivation, social support) that are supposed to be critical in meeting specific internal and external demands. However, the synchronous exposure to diverse chronic stressors may lead to a cumulative depletion of resources and thereby gradually reduce the capabilities of coping with the demands experienced, thereby reinforcing the negative effects on health (Ford et al., 2014).

The relevance of the role that resources play in the development of health issues is underpinned by our findings according to which job resources were significantly related to symptom clusters, even though the effects were found to be less strong than those of the stress profiles. Self-efficacy, however, was only related to the exhausted class, in that higher self-efficacy went along with a decrease in the odds of belonging to this class. Again, this effect was relatively weak. Overall, these findings largely conform to previous studies that found both job and personal resources to be important predictors of health and psychological well-being (Kalimo, Pahkin & Mutanen, 2002; Mayerl et al., 2016).

Strengths and limitations

The methodological approach of our study allowed us (1) to include a variety of different variables in order to cover a broad range of potentially relevant aspects, (2) to take a potential heterogeneity in the population into account, and (3) to examine the effects of and consequences for empirically identified constellations of variables. Combining person- and variable-centred analyses thus has the advantage of capturing phenomena that emerge in varying degrees or only among specific groups of individuals (Muthén & Muthén, 2000). For instance, we found diverse constellations of job demands to be differentially associated with specific symptom clusters. Such a phenomenon would have remained concealed by collapsing several demands and health indicators into one composite score, respectively, and examining their relationship without accounting for unobserved heterogeneities. We thus consider our approach to be an at least equally valid alternative to the purely variable-centred approaches more widely used in work-related stress research (see also Wang & Hanges, 2011).

One limitation can be attributable to the fact that this study was based on data from a cross-sectional design and that the revealed relationships allowed no claims concerning causality. Future research should therefore examine the link between stress profiles and symptom clusters in longitudinal studies as well, in order to investigate developmental pathways in latent class membership for health problems as a function of changing levels of job demands over time. For example, where job demands persist over an extended period of time or when they become more severe and simultaneously resources are not available to cope with these demands, the probability of being classified into more serious symptom clusters may increase (see McEwen, 1998).

A second shortcoming of this study might reside in the fact that most of the instruments used in the survey questionnaire are either adapted/shortened versions of original scales or measures that have not been validated in previous studies. Although all of the measures used in our study demonstrate face validity, this issue may be nevertheless a potential source of bias. For example, it remains unclear how much the shortened version of the General Self-efficacy Scale has in common with the original version of this scale. Furthermore, the measures of job demands, job resources, and health symptoms can be considered only as proxy measures reflecting the underlying concepts relevant for our research questions (for a similar approach see e.g., De Jonge et al., 2000). Similarly, there might be also a bias due to the self-reported character of the data. The inclusion of more objective and standardized variables (e.g., measures based on medical examinations) thus could have further benefited the analysis.

A third critical point concerned the response categories of the indicator variables. Due to the relatively high number of variables, we reduced complexity and facilitated interpretability of the models by dichotomizing all indicator variables before performing LCA. Although this is a common approach in health research, the kind of collapsing strategy considerably affects interpretation of results. This needs to be kept in mind when comparing the estimates of our latent class models with those of previous and future studies.

Finally, the method of classifying individuals according to their most likely class implies that some uncertainty regarding latent class membership always remains. This fact poses statistical problems when using class membership as an observed variable in further analyses (e.g., biased coefficient estimates in regression analysis; Bolck, Croon & Hagenaars, 2004). However, in a simulation study, SL Clark & BO Muthén (2009, unpublished data) found that when entropy was above 0.8, this approach performed better with regard to recovering the true effect in regression analysis than alternative methods. On the other hand, they also argue that standard errors may be underestimated and thus recommend to rely on a more stringent significance criterion than α = 5%. Since both apply to our study (entropy measures greater than 0.8 and a stringent significance criterion of α = 1%), we did not consider this to be an issue in this instance.

Conclusion

This study supports the strengths of a combined person- and variable-centred approach in work-related stress research. Since we found the working population to consist of heterogeneous groups in terms of both the burden perceived from adverse working conditions and the susceptibility to health problems, and found diverse constellations of demands to affect health to different degrees, the current approach permitted a more differentiated and yet holistic view of the stress phenomenon than the use of the more conventional, purely variable-centred methods.

Our results also suggest a hierarchical order for different patterns of job demands in terms of the seriousness of encountered health effects. While a mainly psychosocially burdening working environment seemed to be least severe, it seems important to note that such an adverse environment was most prevalent and was nevertheless strongly associated with ill-health. Workers predominantly burdened by physical demands, however, showed an even greater likelihood of reporting health symptoms while the synchronous exposure to extensive physical, psychosocial, and organizational demands dramatically increased the odds of falling into the most critical symptom cluster. Intervention strategies should thus especially focus on jobs with a high cumulative burden due to several demands, not only to promote health and well-being, but also to prevent serious diseases.

We would like to thank the Upper Austrian Chamber of Labour for granting us access to their data and for their considerate cooperation.

Additional Information and Declarations

Competing Interests

Author Contributions

Human Ethics

Data Availability

The authors declare there are no competing interests.

Hannes Mayerl conceived and designed the experiments, analyzed the data, wrote the paper, prepared figures and/or tables, reviewed drafts of the paper, interpreted the results.

Erwin Stolz, Anja Waxenegger and Wolfgang Freidl conceived and designed the experiments, reviewed drafts of the paper, interpreted the results.

The following information was supplied relating to ethical approvals (i.e., approving body and any reference numbers):

The Ethics Committee of the Medical University of Graz approved this study (Ethical Application Ref: 27-251 ex 14/15).

The following information was supplied regarding data availability:

Open Science Framework: https://osf.io/a76wh/.

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
