# Peer review of "Exploring differential health effects of work stress: a latent class cluster approach"

_PeerJ, doi:10.7717/peerj.3106_

## Round 0.1 · original submission · Major Revisions

· Academic Editor

Major Revisions

I now have received two reviewers' comments. Although both reviewers expressed their interest in your study, several aspects of this manuscript should be revised to improve its clarity. Their observations are presented with clarity so I'll not risk confusing matters by belaboring or reiterating their comments. While I might quibble with the occasional point, I note that I regard the reviewers' opinions as substantive and well-informed. I believe that all of the highlighted reservations require contemplation and appropriate attention in revising the document if it is to contribute appropriately to Peer J and the extant literature. Please revise or refute according to the two reviewers' comments and provide a point by point reply in addition to the revised manuscript.

Tsung-Min Hung, Ph.D.
PeerJ editor
Distinguished professor
Department of Physical Education
National Taiwan Normal University

Reviewer 1 ·

Basic reporting

Generally this manuscript has very good overall quality such as excellent language quality and writing style, thorough literature review, validity of data, appropriateness of data analyses, and completeness of the discussion. There are only few minor questions that might considerably diminish the readability of this paper.

◆In P.2 (L.58, L.69), the authors introduced two related research that used cluster approach. Given that indicators used to distinct the groups is one of the most important information in this type of research, I would suggest a little more detail about the indicators of these two key references. Thus readers could gain more information to know and to interpret the classes in the present study.

◆In P.2 L.74, „this study not only....“. Is „this study“ „Vanroelen et al.“? Or it did mean the present study? As a reader, it really confused me.

◆In P.3 L.119~, the authors proposed the hypothesis that greater job demands would lead to more health symptoms. I'm wondering if it's a proper hypo for the present study. The authors brought up so many reasons that we should consider the heterogeneous of stress and stress-related health problems. And a „the more ... the more“ hypo seems to diminish the value of this issue.

◆In P.3 L.123~, all of a sudden, job resources, self-efficacy, and even another two models (JD-C and ER) burst into this paper. It's not easy to understand the relevance of these concepts. Later in METHODS section, we see these variables were taken into account as covariates? There should be more rationale description in P.3 to make it reasonable.
Besides, P.5 L.189~192, the description here is difficult to understand. If the scales of JD-C and ER models were not utlitized in the original survey, then there's no convincing argument to propose JD-C and ER models.

Experimental design

◆The data were collected over a long span of five years. Is it possible that samples overlap? How would it influence the interpretation of research findings?

◆P.6 L.229~230, is there any methodology reference about the combination of 1~3 and 4~5? The authors did give reasons in Limitations. But it would be better if there's a brief explanation here, or at lease a hint for the readers about where they could find more information.

Validity of the findings

◆P.13 L.398~, here Vanroelen's research findings were introduced in quite detail. However, the paragraph end up without any explanation about the similarity /difference between Vanroelen's study and the present study.

◆(About the limitaton) Health symptoms in the present study were measured with the rating scale „how often“, but not „how severe“ (like quantity v.s. quality). „More symptoms“ is not necessary equal to „severe“ health problems (in L.25, L.121, L.430, L.526). Would it influence the interpretation of research findings?

Reviewer 2 ·

Basic reporting

1) Clear, unambiguous, professional English language used throughout.
Overall this study is precise and free of any grammatical or structural written errors.

2) Intro & background to show context.
The paper clearly introduces the past findings regarding health clusters and stress profiles in different working fields. Based on the previous findings, the authors have properly evidenced that it would be more appropriate to use the latent class analysis to identify different health symptom clusters and stress profiles. Further, this study indicated the importance of estimating the relationships between symptom cluster and stress profile, job resources, and self-efficacy.

3) Literature well referenced & relevant.
Literature has been well-referenced and relevant. The format follows the standard of APA style.

4) Structure conforms to PeerJ standard, discipline norm, or improved for clarity.
The structure of the manuscript conforms to the standard of the PeerJ and the norm of relevant disciplines.

5) Figures are relevant, high quality, well-labelled & described.
Figures and tables are relevant and well-labelled that help readers realize the categories of health symptom clusters and stress profiles, as well as the distribution of the latent class probabilities for the symptom clusters and stress profiles over gender, age, education, and educational groups. The Table 4 of multinomial logistic regression model for health symptoms is also clear.

Experimental design

1) Original primary research within Scope of the journal.
This study is well-focused and properly builds on previous work. The purpose of this study falls within the scope of the journal, which is to examine the effects of various constellations of job demands on health symptom clusters.

2) Research question well defined, relevant & meaningful. It is stated how research fills an identified knowledge gap.
The description of research question is well-stated that makes the potential of this study to bring new relevant information to the field clear at this point. Based on the previous findings, the authors have criticized the existing knowledge gaps and then pointed out how this study can solve these questions.

3) Rigorous investigation performed to a high technical & ethical standard.
This study is well designed and also follows the ethical standard to claim the approval of the Ethics Committee while recruiting the working population at various ages. Trained interviewers administered structured face-to-face interviews and informed participants on the confidentiality and anonymity of the collected data.

4) Methods described with sufficient detail & information to replicate.
The materials and methods described in the manuscript are moderately sufficient for other researchers to replicate. However, the authors are recommended to explain how they decided to include 16 items as health symptoms, and 19 items as job demands. It will be reasonable to provide evidence to support the validity of these two measures. The measure of self-efficacy is usually situation- specific or task-specific. Is it supposed to use a job-related self-efficacy measure in this study? Why did the authors utilize a measure of general self-efficacy?

Validity of the findings

1) Negative/inconclusive results accepted.
In general, all relationships among stress profiles and health symptom clusters were reasonable, which were mostly consistent with the previous findings and the directions of the predictions.

2) Data is robust, statistically sound, & controlled.
The data provided is robust enough because the multinomial logistic regression model could control the influences of age, gender, job resources, education, and occupations of the participants. This study used a more precise technique, the latent class analysis, to classify the categories of health symptoms and stress profiles. Regarding statistical techniques, this study utilized a reliable method to examine the research questions.

4) Conclusion well stated, linked to original research question & limited to supporting results.
The conclusion is appropriately stated based on the results of this study and the findings of previous work. Also, the authors have raised their arguments, limitations, and recommendations according to the findings of this study.

Additional comments

Generally, the justification of research questions is clear which makes this study contribute to the field regarding work stress and health concerns. The authors are suggested to clarify the validity of the measures regarding health symptoms, job demands, and self-efficacy. Then, the validity of these measures will be improved, which can increase the contribution of the findings. This study will be able to provide new information regarding the relationships of job demands and health symptom clusters.

---

## Round 0.2 · accepted · Accept

· Academic Editor

Accept

I have now received two reviewers’ comment and both reviewers were satisfied with your reply and revisions from previous comments. You and your coauthors have my congratulations. Thank you for choosing PeerJ as a venue for publishing your research work and I look forward to receiving more of your work in the future.

Tsung-Min Hung, Ph.D.
PeerJ editor
Distinguished professor
Department of Physical Education
National Taiwan Normal University

Reviewer 1 ·

Basic reporting

The comments were addressed detailedly and clearly. And throughout the manuscript, all the contents are described appropriately and I believe the article will be of interest to readers of this journal and related research areas.
Only two small questions.
1. please make sure that if „0“ is needed or unnecessary when the Cronbach's α is presented. Like Cronbach's α = .88 (L.225)
2. the significant sign „*“ in table 2 & table 3 was marked right after the standardized residuals. Would it be more accurate to mark it after the Mean (or the percentage in the present study)?

Experimental design

no comment

Validity of the findings

no comment

Additional comments

no comment

Reviewer 2 ·

Basic reporting

No comments.

Experimental design

The authors have appropriately clarified the validity of the measures regarding health symptoms, job demands, and self-efficacy, and added further information to the manuscript. The validities of these measures have been well described, which can increase the contribution of the findings of this study.

Validity of the findings

No comments.

Additional comments

These revisions can improve the quality of the manuscript and bring new information regarding the relationships of job demands and health symptom clusters.